# Validation of the Champion Health Belief Model Scale for an Investigation of Breast Cancer Screening Behaviour in Malaysia

**DOI:** 10.3390/ijerph18179311

**Published:** 2021-09-03

**Authors:** Mila Nu Nu Htay, Désirée Schliemann, Maznah Dahlui, Christopher R. Cardwell, Siew Yim Loh, Nor Saleha Binti Ibrahim Tamin, Saunthari Somasundaram, Victoria Champion, Michael Donnelly, Tin Tin Su

**Affiliations:** 1Centre of Population Health, Department of Social and Preventive Medicine, Faculty of Medicine, University of Malaya, Kuala Lumpur 50603, Malaysia; drmlnnh@gmail.com (M.N.N.H.); maznahd@ummc.edu.my (M.D.); 2Department of Community Medicine, Melaka-Manipal Medical College, Manipal Academy of Higher Education (MAHE), Melaka 75150, Malaysia; 3Centre for Public Health and UKCRC Centre of Excellence for Public Health, Queen’s University Belfast, Belfast BT12 6BA, UK; D.Schliemann@qub.ac.uk (D.S.); c.cardwell@qub.ac.uk (C.R.C.); michael.donnelly@qub.ac.uk (M.D.); 4Department of Health Policy and Administration, Faculty of Public Health, Universitas Airlangga, Surabaya 60115, Indonesia; 5Department of Rehabilitation Medicine, University of Malaya, Kuala Lumpur 50603, Malaysia; syloh@um.edu.my; 6Ministry of Health, Putrajaya 62590, Malaysia; drnorsaleha@moh.gov.my; 7National Cancer Society of Malaysia, Kuala Lumpur 50300, Malaysia; saun@cancer.org.my; 8School of Nursing, Indiana University, Indianapolis, IN 46202, USA; vchampio@iu.edu; 9South East Asia Community Observatory (SEACO), Jeffery Cheah School of Medicine and Health Sciences, Monash University Malaysia, Bandar Sunway 47500, Malaysia

**Keywords:** health beliefs, Champion Health Belief Model Scale, breast cancer, mammogram, Malaysia, translation, validity, reliability

## Abstract

Breast cancer (BC) is the most common cancer among women globally, including in Malaysia. There is a need to assess women’s beliefs about BC and screening in different cultural settings. This study aimed to translate and validate an adapted version of the United States (US) Champion Health Belief Model Scale (CHBMS) for an investigation of predictors of BC screening in Malaysia. The CHBMS was adapted, and forward and backward translated into the Malay language. The validity and reliability of the CHBMS-BC-M (M for Malay language) was investigated in a community sample of 251 multi-ethnic Malay-speaking women. Principal component analysis with varimax rotation indicated that the structure of the adapted CHBMS-BC-M comprised three subscales with 21 items, and an Item-Content Validity Index (I-CVI) of 0.83 and above for all items. The explanatory factor analysis (EFA) showed acceptable to high factor loadings on items. Cronbach’s alpha for the subscales ranged from 0.59 to 0.87. The reproducibility of the scale was fair to high, with an Intraclass Correlation Coefficient (ICC) of 0.53 to 0.80 for the subscales. Overall, the analysis indicated that the translated CHBMS-BC-M is a valid and reliable scale to measure beliefs about BC and screening in the Malay-speaking ethnic population of Malaysia.

## 1. Introduction

Globally, breast cancer (BC) is the most commonly diagnosed cancer with 24.2% of all new cancer cases [1], and it is the leading cause of cancer mortality among women worldwide [2]. Similarly, BC is the most common female cancer in Malaysia and accounts for 34.1% of all cancers among females [3]. Early diagnosis and management are crucial to improving survival rates. The Ministry of Health Malaysia offers opportunistic screening in the form of Clinical Breast Examinations (CBE) and mammograms for eligible women. However, the presentation of BC is often late in Malaysia (47.9% present at stage 3 or 4) [3]. Moreover, a systematic review reported that the uptake of mammograms is low and varied from 3.6% to 30.9% in the general population in Malaysia [4].

The Health Belief Model (HBM) offers a conceptualisation or way of understanding the relationship between an individual’s beliefs and their health behaviours [5]. Because mammography uptake is low, there is a need to investigate beliefs and perceptions about BC and BC screening in order to identify potential barriers to screening uptake in Malaysia [6]; it is likely that the views of women in Malaysia and South East Asia differ from the beliefs and perceptions of women in Western countries. A better understanding of the beliefs that are held by the local population would aid the development of strategies designed to increase BC screening uptake. Initially, the Champion’s Health Belief Model Scale (CHBMS) was developed to explain breast self-examination (BSE) behaviour in the United States (US), which included subscales of perceived susceptibility, seriousness, benefits, barriers, and health motivation [7]. In 1993, the confidence subscale for BSE among women was added to the CHBMS [8]. The CHBMS was also revised in 1999 to take the increasing use of mammogram screening into account [9]. The final revised version of the CHBMS-BC consists of three subscales, i.e., perceived susceptibility, perceived benefits and perceived barriers, and has been used to explain the utilisation of mammogram screening [9]. Theoretically, it is expected that women who report a higher perceived susceptibility to BC and perceived benefits to screening will be more likely to attend screening programmes (compared to women who perceive more barriers to screening) [9].

The CHBMS for BSE was translated and validated into the Korean [10], Arabic [11], Turkish [12], and Indonesian language [13]. According to these previous studies, the longer version of the CHBMS for BSE resonates well in different languages [10,11,12,13]. The first Malay version of the CHBMS included the benefits and barriers subscales for BSE, CBE, and mammograms. It comprised 10 subscales and 63 items and the majority of respondents represented the Malay ethnic group (84.4%) [14]. 

Current BC screening guidelines in Malaysia recommend CBE and mammogram screening for the general population [15]. BSE is used to raise breast health awareness rather than serve as a mode of screening [16]. Since Malaysia is a multi-ethnic country, it is important to assess the validity of the CHBMS-BC across the three major ethnic groups. The purpose of this study was to adapt, translate, and validate the revised version of the CHBMS-BC [9] in order to explain BC screening behaviour among multi-ethnic Malaysian women. The research questions and hypotheses of the study were:i.Are the items of the CHBMS-BC-M relevant and culturally appropriate to measure the health beliefs about BC screening among women in Malaysia?ii.Are the items of the CHBMS-BC-M valid to measure the health beliefs about BC screening among women in Malaysia?iii.Are the subscales of the CHBMS-BC-M reliable to measure the health beliefs about BC screening among women in Malaysia?

**Hypothesis** **1** **(content** **validation).***Individual items of the CHBMS-BC-M are relevant, and culturally appropriate to measure the health beliefs about BC screening*.

**Hypothesis** **2** **(construct** **validation).***The findings of this study confirm that the proposed three subscales of the CHBMS-BC-M are valid to measure the health beliefs about BC screening*.

**Hypothesis** **3** **(reliability).***The subscales of the CHBMS-BC-M show moderate to high internal consistency and test–retest reliability*.

## 2. Materials and Methods

The steps describing the CHBMS-BC-M validation are summarised below. This study received ethical approval from the University of Malaya Medical Centre (UMMC) Ethics Committee (Ref No. 2016126-4668) and Melaka-Manipal Medical College Research ethical committee.

The steps of the adaptation, translation, and validation process are outlined in Figure 1.

### 2.1. Adaptation of Measurement Tool

The revised version of the CHBMS-BC (English version) consists of three subscales and 19 items: perceived susceptibility (*n* = 3), perceived benefits (*n* = 5), and perceived barriers (*n* = 11) about mammogram screening [9]. This revised version of the CHBMS-BC (English version) was modified, translated into Malay, and validated. Permission to validate the scale for use in Malaysia was obtained from the research team who designed the original scale.

Since BC screening programmes in Malaysia focus on CBE and mammogram screening [15], ‘mammogram screening’ was changed to ‘BC screening’. An expert panel, consisting of public health specialists (*n* = 2), an expert in instrument development (*n* = 1), and translation and language specialists (*n* = 2), reviewed the scale critically and made recommendations about the modification of its content and the adaptation of its response format with the aim of increasing the cultural sensitivity and appropriateness of the scale for use by Malaysians and users of the local health care system. Three items were added to the barriers subscale: (i) I cannot afford to get BC screening, (ii) I don’t receive the encouragement from my close relatives that I need to attend BC screening, and (iii) I am afraid that a male doctor will carry out the BC screening. The initial validation of the CHBMS-BC-M included 22 items and three subscales, susceptibility (*n* = 3), benefits (*n* = 5), and barriers (*n* = 14). The response to each item was recorded on a five-point Likert scale that ranged from strongly disagree (1) to strongly agree (5).

### 2.2. Validation Procedure

#### 2.2.1. Linguistic Validation

There are three major ethnic groups in Malaysia, namely Malay (native Malaysian), Chinese Malaysian, and Indian Malaysian. However, the Malay language is the national and official language, and it is mainly used in daily communication. Therefore, the CHBMS-BC-M was translated and validated into the Malay language. The CHBMS-BC-M was forward and backward translated to the Malay language by two independent bilingual experts. The initial translated Malay version was reviewed by five bilingual experts who checked the accuracy and conceptual equivalence with the English language version. A barrier item, ‘Saya tidak berkemampuan untuk pemeriksaan payudara’ was amended by the experts to read, ‘Saya tidak berkemampuan untuk mendapatkan pemeriksaan kanser payudara’ (I cannot afford to get breast cancer screening) in order to retain the same conceptual meaning as the original English language version. Any inconsistencies were then discussed, and the accuracy and adequacy of the CHBMS-BC-M were ascertained via backward translation to English. The backward translated items and the original items were consistent in their meaning and concepts.

#### 2.2.2. Content Validation

Six bilingual speakers (a cancer advocate, an oncologists, a psychiatrist, and three public health medicine specialists) rated the relevancy of the items.

#### 2.2.3. Pilot Testing

During the pilot testing, respondents from the three main ethnic groups were included (10 Malay, 10 Chinese and 10 Indian). Respondents were aged 40 years and above and were recruited from the general population to identify the possible minor differences between the different ethnic groups such as the ambiguity, clarity of the questions and any disagreeable items.

#### 2.2.4. Construct Validation

The further validation of the CHBMS-BC-M was carried out in Melaka, the fifth largest city, located in the southern region of Malaysia. Data collection was carried out between April and May 2018. Women were eligible to participate in the survey if they were aged 40 years and above, spoke Malay, could answer questions independently and lived in the study area. Women with a previous or current diagnosis of BC as well as working health care professionals (e.g., doctors, nurses, and allied health care professionals) were excluded from participation. All respondents were informed about the purpose of the study and provided written informed consent.

The sample size was calculated based on the guideline for a meaningful factor analysis that requires ten respondents for each item [17]. Therefore, a minimum of 220 respondents was required for the validation of the 22-item CHBM-BC-M. To obtain respondents from the three major ethnic groups, the quota sampling method was used. In total, 251 women were interviewed: Malay (33.9%), Chinese (33.5%), and Indian (32.7%).

Exploratory Factor Analysis (EFA) was carried out to ascertain construct validity.

#### 2.2.5. Internal Consistency Reliability

The internal consistency reliability of the CHBMS-BC-M was ascertained by analysing Cronbach’s alpha.

#### 2.2.6. Test–Retest Reliability

Fifty respondents from the total sample (*n* = 251) were selected randomly and interviewed twice (with a two-week interval) to test the reproducibility or reliability of the CHBMS-BC-M.

### 2.3. Statistical Analysis

#### 2.3.1. Content Validation

The Item-Content Validity Index (I-CVI) was calculated based on the ratings that were provided by the content validation panel (n=6). Therefore, items with I-CVI of >0.83 were included in the CHBMS-BC-M [18,19].

#### 2.3.2. Construct Validation

The response to each item was recorded on a five-point Likert scale. The mean score and standard deviation (SD) were calculated for each subscale. Construct validation was tested by conducting an EFA (*n* = 251) that used principal component analysis with varimax rotation and the PASW Statistics for Windows, Version 18.0 (SPSS Inc., Chicago, IL, USA).

#### 2.3.3. Internal Consistency Reliability

Cronbach’s alpha was calculated using the answers from the community survey (*n* = 251) to assess the internal consistency of the items in each subscale. It was deemed that an item should be removed from the scale if the corrected item-total correlation was <0.3 [20,21]. We retained the important items that reflect the theoretical domain of the scale, which has been recommended by Rattray and Jones [22].

#### 2.3.4. Test–Retest Reliability

The intra-class correlation coefficient (ICC) was calculated to assess the test–retest reliability for the CHBMS-BC-M.

Mean scores of the CHBMS-BC-M subscales were compared between women who reported to have previously participated in BC screening (CBE or mammogram) and those who did not by using the independent sample *t*-test.

## 3. Results

### 3.1. Content Validation

The scores for the Item-Content Validity Index (I-CVI), based on the judges’ ratings, were 0.83 and above and, therefore, the 22 items were retained in the CHBMS-BC-M to proceed with the construct validation.

### 3.2. Pilot Testing

Women (*n* = 30) responded that the questions or items were clear and easy to understand. A degree of hesitancy among the sample of women in responding to the three items in the susceptibility subscale was observed by the research interviewer, perhaps because these items address a participant’s perceived susceptibility to developing BC. The hesitant behaviour was not deemed to be sufficiently significant to justify the removal of the items.

### 3.3. Socio-Demographic Characteristics of Respondents

The sample (*n* = 251) comprised equal proportions of Malay (33.9%), Chinese (33.5%) and Indian women (32.6%). About half of all respondents completed secondary school education (Table 1). Eighty-two percent of respondents were married and approximately 55% were employed.

### 3.4. Construct Validation

The EFA applied principal component analysis with varimax rotation to the data set (which comprised the responses to the CHBMS-BC-M from 251 women) in order to determine the dimensionality of the 22-item scale. All items had a factor loading of 0.4 and above except for the benefits subscale item no. 3 (BE 3). Therefore, item BE 3 was removed from the scale. The Kaiser–Meyer–Olkin (KMO) measure of sampling adequacy was 0.82, indicating that the sample size was adequate to proceed with the factor analysis [23]. Bartlett’s Test of sphericity was 1701.195 with a degree of freedom of 210 and a level of significance of <0.001. The structure of the CHBMS-BC-M obtained 21 items, which loaded in three subscales and explained 38.86% of the total variance. The percentages of variance explained by the subscales in the factor analysis are reported in Appendix A. Items loaded in their respective subscale and did not cross-load at >0.3 on other subscales (Table 2).

### 3.5. Internal Consistency Reliability

The three items in the perceived susceptibility subscale had a Cronbach’s alpha of 0.84, which indicated high reliability of the subscale [24]. The corrected item-total correlation was 0.68 and above. Four items were included in the perceived benefits subscale and the Cronbach’s alpha value was 0.59 which was a moderately reliable subscale [24]. The perceived barriers subscale included fourteen items and the Cronbach’s alpha was 0.87, which is considered as a highly reliable subscale [24] (Table 3).

### 3.6. Test–Retest Reliability

The intra-class correlation coefficient (ICC) for the susceptibility subscale was 0.63 (good), benefits subscale was 0.53 (fair) and barriers subscale was 0.80 (excellent).

### 3.7. Association between CHBM-M and Breast Cancer Screening Uptake

Women who had received BC screening in the past (CBE, *n* = 112; mammogram, *n* = 86) rated the ‘perceived benefits’ significantly higher and ‘perceived barriers’ significantly lower compared to women who had not experienced screening (CBE, *n* = 139; mammogram, *n* = 165). The mean scores for ‘perceived susceptibility’ were similar in both groups (Table 4).

## 4. Discussion

In this study, the revised version of CHBMS-BC with three subscales was adapted, translated into Malay, and validated. The findings indicated that the CHBMS-BC-M is an acceptable, reliable, and valid tool to measure the health beliefs of women regarding BC screening in Malaysia. The longer CHBMS (comprising six subscales) for BSE was translated and validated in Indonesia [13], Malaysia [14], Turkey [12], Korea [10], and Arabic countries [11]. However, to our knowledge, this study was the first to translate and validate the revised shorter version of the CHBMS-BC [9] with three subscales.

Since Malaysia is a multi-ethnic country, perception of health, illness and healthcare-seeking behaviours are influenced by peoples’ culture, beliefs, and traditions [25,26,27]. To determine the validity and reliability of the CHBMS-BC-M across the different ethnic groups, this study was conducted with respondents from the three major ethnic groups in Malaysia.

In the factor analysis, the correlation of each item with the respective subscales revealed that all items had factor loadings of 0.4 and above. Similar factor loadings (0.40–0.90) were reported in the English version of the CHBMS-BC [9]. Only one item did not perform well, i.e., “If I find a lump through breast cancer screening, my treatment for breast cancer may not be as bad’, and the item was removed from the scale. This result may reflect a degree of uncertainty among women regarding the content of this item and there may be a need to give attention to ways in which to increase knowledge about, for example, BC screening (either CBE or mammogram), the different stages of BC and its treatment. Generally, the analysis indicated that the final CHBMS-BC-M included 21 items and resonated well with women in Malaysia.

The Cronbach’s alpha coefficient of the three subscales indicated that the internal consistency reliability was moderate for the benefits subscale and high for the susceptibility and barriers subscales. This is similar to the English version of the revised CHBMS-BC [9], except for the benefits subscale, which was lower than the benefits subscale of revised CHBMS-BC [9] but acceptable Cronbach’s alpha.

According to Cicchetti (1994), an ICC in the range of 0.4–0.59 is considered fair, 0.6–0.74 as good, and 0.75–1.00 as excellent [28]. Therefore, the susceptibility subscale had a good ICC, the benefits subscale had a fair ICC, and the barriers subscale had an excellent ICC for the test–retest reliability. These findings suggest that the CHBMS-BC-M may be considered to be a stable scale for measuring the health beliefs of Malaysian women.

The past history of BC screening was self-reported by the respondents and those who had “CBE within the last one year” or “mammogram within the last two years” were considered as women who participated in BC screening. Previous studies in Malaysia suggested fear and denial of cancer diagnosis, fatalism, and beliefs that life events are predetermined and inevitable among Malaysian women [29,30]. Many respondents in our study hesitated to answer the items in the susceptibility scale—beliefs or attitudes such as fear, denial, and accepting cancer as a predetermined life event might explain the apparent similarity regarding scores on the susceptibility subscale between women who participated in BC screening and who did not participate. Malaysian women who utilised BC screening programs scored higher on the benefits subscale compared to women who did not take up screening opportunities. The barriers subscale included an additional three items regarding finance, family encouragement, and examination by a male doctor in an attempt to capture potential local cultural concerns. The mean scores for the barriers subscale differed significantly between women who screened and did not screen. Similar findings for the benefits and barriers subscales were observed in the US CHBMS-BC validation study [9]. Women who never had a mammogram reported lower perceived benefits and higher perceived barriers scores compared to those who had a mammogram within the last 15 months [9].

A significant benefit of this shorter validated version of the CHBMS-BC-M is that it is less time-consuming and less demanding for respondents. The validated brief scale is likely to improve completion and response rates. It may be used in routine clinical and community settings and responses on the scale may be used by clinicians and public health practitioners to guide advice-giving and screening behaviour according to reported perceived barriers, susceptibility and benefits to screening [9]. Ideally, the CHBMS-BC-M should be tested in further studies with different population groups in order to learn further about the performance of the scale and improve its generalisability for Malaysia.

### Strengths and Limitations

The strength of this study was that a meticulous validation process was carried out for the CHBMS-BC-M to measure the health beliefs about BC and screening among women. The cultural adaptation and validation were tested among the three major ethnic groups and; therefore, the scale is applicable across multi-ethnic Malay-speaking women in Malaysia.

This study had some limitations. The non-probability sampling (quota sampling) might impose selection bias and might limit the generalizability of the findings. The benefits subscale has a lower Cronbach’s alpah coefficient compared to the other two subscales. Therefore, further refinement of the scale with a wider population would be beneficial.

## 5. Conclusions

The Malay 21-items CHBMS-BC-M is generally found to be a valid and reliable tool to measure the beliefs of multi-ethnic Malay-speaking Malaysian women regarding BC screening. Further refinement and testing of the items may be needed in a larger population. This study contributed a tool for the assessment of the perceived susceptibility, benefits and barriers of women regarding BC and BC screening in the Malay language. This tool is applicable to access women’s health beliefs in both clinical settings and in the community. In addition, the CHBMS-BC-M may be used as an evaluation tool to assess changes in women’s health beliefs before and after BC awareness-raising interventions in the future.

## Figures and Tables

**Figure 1 ijerph-18-09311-f001:**
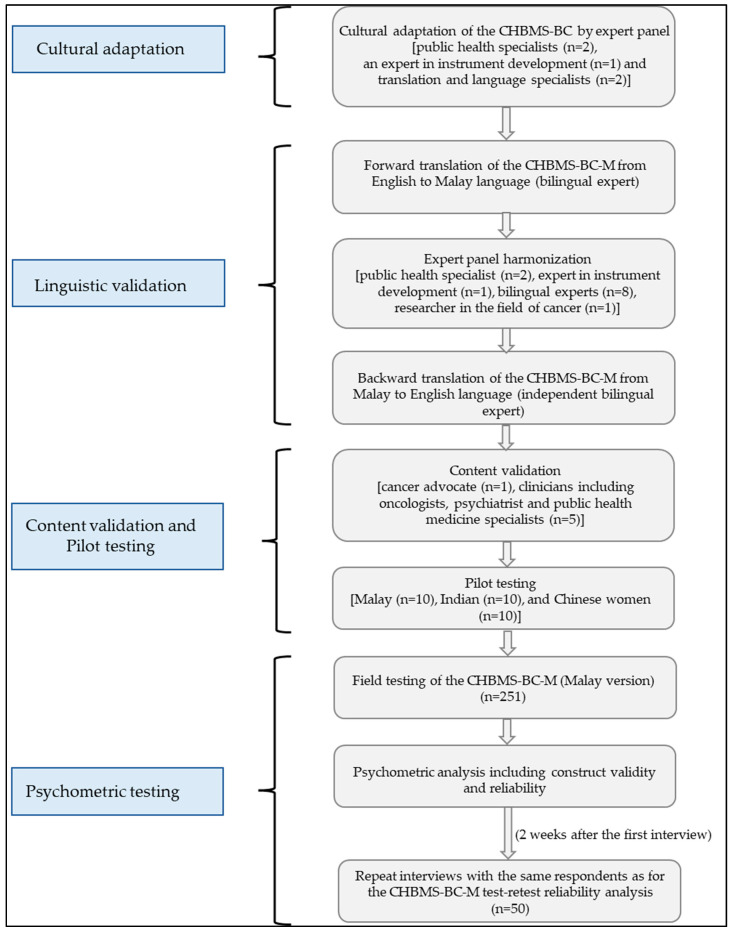
Cultural adaptation, translation, and validation of CHBMS-BC-M.

**Table 1 ijerph-18-09311-t001:** Demographic Patterns of Participants (*n* = 251).

Variables	*n*	%
**Ethnicity**		
Malay	85	33.9
Chinese	84	33.5
Indian	82	32.6
**Religion**		
Islam	86	34.3
Hinduism	75	29.9
Buddhism	74	29.5
Others	16	6.3
**Marital Status**		
Married	208	82.9
Single ^1^	42	16.7
No reply	1	0.4
**Number of household members**		
1–5 members	169	67.3
6–10 members	78	31.1
More than 10 members	1	0.4
No reply	3	1.2
**Educational level**		
No formal education	7	2.8
Primary education	56	22.3
Secondary education	123	49.0
Tertiary education	61	24.3
Others	4	1.6
**Current Job Status ^2^**		
Employed	137	54.6
Unemployed	113	45.0
No reply	1	0.4
**Monthly family income ^3^**		
Below RM 3000	137	54.6
RM 3000–RM 5000	58	23.1
Above RM 5000	46	18.3
No reply	10	4.0

^1^ Single – includes participants who were widowed, divorced and who never married. ^2^ Employed—includes civil servant, private sector employee, and self-employed. Unemployed—includes government retiree, private retiree, and home maker. ^3^ Income of all household members combined.

**Table 2 ijerph-18-09311-t002:** Factor loadings of items in the CHBMS-BC-M.

No	Item	Factor Loadings
**Susceptibility**		
1	Saya bekermungkinan akan mendapat kanser payudara. (It is likely that I will get breast cancer.)	0.87
2	Kemungkinan saya mendapat kanser payudara dalam beberapa tahun ini adalah tinggi. (My chances of getting breast cancer in the next few years are high.)	0.86
3	Saya merasakan saya akan mendapat kanser payudara pada bila-bila masa sepanjang hayat hidup saya. (I feel I will get breast cancer sometime during my life.)	0.86
**Benefits**		
1	Sekiranya saya mendapat pemeriksaan kanser payudara dan tiada apa dijumpai, saya tidak perlu risau tentang kanser payudara. (If I get screened for breast cancer and nothing is found, I don’t need to worry as much about breast cancer.)	0.54
2	Pemeriksaan kanser payudara akan membantu saya mencari ketulan di payudara lebih awal. (Having breast cancer screening will help me find breast lumps early.)	0.67
3	Pemeriksaan kanser payudara adalah cara terbaik bagi saya untuk mengesan ketulan yang amat kecil. (Having breast cancer screening is the best way for me to find a very small lump.)	0.79
4	Dengan menjalani pemeriksaan kanser payudara, kemungkinan untuk saya mengalami kematian akibat kanser payudara akan berkurang. (Having breast cancer screening will decrease my chances of dying from breast cancer.)	0.77
**Barriers**		
1	Saya bimbang untuk menjalani pemeriksaan kanser payudara kerana saya mungkin mengetahui sesuatu yang tidak kena dengan payudara saya. (I am afraid to have breast cancer screening because I might find out something is wrong.)	0.68
2	Saya takut menjalani pemeriksaan kanser payudara kerana saya tidak faham apa yang akan dilakukan. (I am afraid to have breast cancer screening because I don’t understand what will be done.)	0.63
3	Saya tidak tahu bagaimana untuk mendapatkan pemeriksaan kanser payudara. (I don’t know how to go about getting breast cancer screening.)	0.62
4	Saringan pemeriksaan kanser payudara adalah sangat memalukan. (Having breast cancer screening is too embarrassing.)	0.57
5	Pemeriksaan kanser payudara mengambil masa yang agak lama. (Having breast cancer screening takes too much time.)	0.71
6	Pemeriksaan kanser payudara (mammogram) sangat menyakitkan. (Having breast cancer screening (mammogram) is too painful.)	0.69
7	Kakitangan kesihatan adalah kasar sewaktu menjalankan pemeriksaan kanser payudara. (People doing breast cancer screening are rough to women.)	0.58
8	Pemeriksaan kanser payudara (mammogram), mendedahkan saya pada radiasi yang tidak perlu. (Having breast cancer screening (mammogram) exposes me to unnecessary radiation.)	0.49
9	Saya tidak ingat untuk berjumpa doktor untuk mendapatkan pemeriksaan kanser payudara. (I cannot remember to go to the doctor to get breast cancer screening.)	0.54
10	Saya mempunyai masalah lain yang lebih penting berbanding dengan melakukan pemeriksaan kanser payudara. (I have other problems more important than getting breast cancer screening.)	0.63
11	Saya tidak termasuk dalam kumpulan peringkat umur yang memerlukan rutin pemeriksaan kanser payudara. (I am too not the right age to need a routine breast cancer screening.)	0.54
12	Saya tidak berkemampuan untuk mendapatkan pemeriksaan kanser payudara. (I cannot afford to get breast cancer screening.)	0.60
13	Saya tidak mendapat galakan yang diperlukan dari saudara terdekat untuk menjalani pemeriksaan kanser payudara. (I don’t have the encouragement I need from my close relatives to attend breast cancer screening.)	0.57
14	Saya merasa bimbang dengan kemungkinan seorang doktor lelaki yang akan memeriksa payudara saya. (I am afraid that a male doctor will carry out the breast cancer screening.)	0.60

**Table 3 ijerph-18-09311-t003:** Internal consistency (Cronbach’s Alpha) of subscales in the CHBMS-BC-M.

Subscale	Item Number	Corrected Item Total Correlation	Cronbach’s Alpha for the Subscale
Susceptibility	S1	0.71	
	S2	0.68	
	S3	0.71	0.84
Benefits	BE1	0.30	
	BE2	0.39	
	BE3	0.51	
	BE4	0.47	0.59
Barriers	BA1	0.61	
	BA2	0.56	
	BA3	0.53	
	BA4	0.50	
	BA5	0.64	
	BA6	0.61	
	BA7	0.52	
	BA8	0.41	
	BA9	0.46	
	BA10	0.54	
	BA11	0.47	
	BA12	0.51	
	BA13	0.48	
	BA14	0.51	0.87

**Table 4 ijerph-18-09311-t004:** Association between mean scores of the CHBMS-BC-M and breast cancer screening uptake (*n* = 251).

	CBE (Yes)		CBE (No)		Difference in Mean (95% CI)	*p* ^1^	Mammogram (Yes)		Mammogram (No)		Difference in Mean (95% CI)	*p* ^1^
	n	Mean (SD)	n	Mean (SD)			n	Mean (SD)	n	Mean (SD)		
Susceptibility	112	7.65 (2.13)	139	7.62 (1.79)	0.03 (−0.45, 0.52)	0.89	86	7.56 (2.34)	165	7.67 (1.70)	−0.12 (−0.62, 0.40)	0.66
Benefits	112	15.79 (1.90)	139	15.18 (1.77)	0.61 (0.15, 1.06)	0.01	86	15.84 (1.83)	165	15.25 (1.83)	0.59 (0.11, 1.1)	0.02
Barriers	112	29.07 (6.72)	139	35.76 (6.39)	−6.69 (−8.33, −5.06)	<0.001	86	29.38 (7.09)	165	34.55 (6.83)	−5.16 (−6.98, −3.35)	<0.001

^1^ Independent sample *t*-test.

## Data Availability

The data presented in this study are available on request from the corresponding author.

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
