# Peer review of "Validation of the Champion Health Belief Model Scale for an Investigation of Breast Cancer Screening Behaviour in Malaysia"

_ijerph, 2021, doi:10.3390/ijerph18179311_

Round 1
Reviewer 1 Report
The study addresses a need to investigate local beliefs and perceptions about breast cancer screening in Malaysia and develop strategies that increase its uptake. The research, well-designed and meticulously conducted, is relevant to the broad audience of healthcare professionals that reads the International Journal of Environmental Research and Public Health. I have just a few comments:
- On page 6, line 210 the authors note that the structure of the scale “obtained 21 items, which loaded in three subscales and explained 39% of the total variance.” I understand this to mean the analysis produced a three-factor solution. It would be helpful if the eigenvalues as well as the percent of variance explained by each component were also included.
- Cronbach’s alpha for the combined items of the perceived benefits subscale indicated it was less reliable than the other subscales. Did the authors examine the effect of eliminating item BE1 to increase the subscale’s internal consistency?
- The paragraph on page 9, lines 234-237 seems to be a typographical error.
- On page 10, line 300 the authors note that “Equal number of respondents from three major ethnic group could enhance the applicability of the scale across the multi-ethnic Malay-speaking Malaysian women.” However, the scale was tested on a sample that was equally divided between the major ethnic groups so the sentence is confusing. Also, the strengths and limitations section should include a limitation of the study.
- Grammar could be improved.
Overall, the manuscript requires some revision but I recommend it be accepted for publication in the International Journal of Environmental Research and Public Health once the above-mentioned issues have been addressed.
Author Response
- On page 6, line 210 the authors note that the structure of the scale “obtained 21 items, which loaded in three subscales and explained 39% of the total variance.” I understand this to mean the analysis produced a three-factor solution. It would be helpful if the eigenvalues as well as the percent of variance explained by each component were also included.
Thank you very much for the valuable suggestions. We have included the eigenvalues and percentage of variance for each subscale in the appendix table 1. The percentage of variance explained by each component is 22.08, 9.21 and 7.57 respectively.
- Cronbach’s alpha for the combined items of the perceived benefits subscale indicated it was less reliable than the other subscales. Did the authors examine the effect of eliminating item BE1 to increase the subscale’s internal consistency?
Thank you very much for the suggestion. Removing the item (BE1) does not improve the alpha coefficient (Cronbach’s alpha) of more than 1 for benefit subscale. In the original English version of CHBMS, similar findings were reported that benefits subscale Cronbach's alpha value is lower compared to susceptibility and barriers subscales (Champion V, 1999). Therefore, we would like to maintain four items in benefits subscale.
- The paragraph on page 9, lines 234-237 seems to be a typographical error.
Duly noted, Sir/ Madam. Thank you very much for pointing out the typo error. We have removed the sentences.
- On page 10, line 300 the authors note that “Equal number of respondents from three major ethnic group could enhance the applicability of the scale across the multi-ethnic Malay-speaking Malaysian women.” However, the scale was tested on a sample that was equally divided between the major ethnic groups so the sentence is confusing. Also, the strengths and limitations section should include a limitation of the study.
We have corrected the sentence on page 10, line 300. We have added the strengths and limitations in the manuscript. Thank you very much.
- Grammar could be improved.
Thank you very much. We have checked and corrected the grammar throughout the manuscript.
Overall, the manuscript requires some revision but I recommend it be accepted for publication in the International Journal of Environmental Research and Public Health once the above-mentioned issues have been addressed.
Reviewer 2 Report
I congratulate the authors on their work. I have no major comments, but I would like to refer to the formal side.
1. Introduction does not contain typical elements for a scientific article. The authors have not formulated objectives and hypotheses.
2. the article is insufficiently grounded in the current state of knowledge. There is a lack of reliable review of literature and studies on similar topics.
3. Conclusions are very short and perfunctory. They should be expanded.
Author Response
- Introduction does not contain typical elements for a scientific article. The authors have not formulated objectives and hypotheses.
Thank you very much for the suggestions. We have amended the introduction and added the research objectives and hypotheses.
- the article is insufficiently grounded in the current state of knowledge. There is a lack of reliable review of literature and studies on similar topics.
Duly noted, we have included the validation studies on the same topic about CHBMS in the literature review. I would like to explain that the previously conducted validation studies were based on the longer version of CHBMS (1993 version). To the best of our knowledge, our study is the first study in which adaptation, translation and validation was carried out for the shorter (revised) version of CHBMS (1999 version). Therefore, we are not able to include the similar validation studies on the shorter version of CHBMS.
- Conclusions are very short and perfunctory. They should be expanded.
We have expanded the conclusion. Thank you very much.